# Development of a German Physical Literacy Assessment for Children in the Context of Health Promotion—An Explorative Approach

**DOI:** 10.3390/children9121908

**Published:** 2022-12-06

**Authors:** Leonie Krenz, Martin Grauduszus, Marlen Klaudius, Isabel Stolz, Stefanie Wessely, Christine Joisten

**Affiliations:** Department for Physical Activity in Public Health, Institute of Movement and Neurosciences, German Sport University Cologne, Am Sportpark Müngersdorf 6, 50933 Cologne, Germany

**Keywords:** physical literacy, children and adolescents, health literacy, physical activity, socioeconomic status

## Abstract

Addressing physical literacy (PL) has become increasingly relevant to counteract sedentary behaviour in children and youth. We developed an assessment tool to measure and evaluate the promotion of individual PL and its four subdomains: motivation and confidence (the affective dimension), physical competence (physical dimension), knowledge and understanding (cognitive dimension), and participation in physical activity. Within this cross-sectional study, we tested 567 children aged 6–12 years from four primary schools that differed in socioeconomic status (SES). A subsequent exploratory factor analysis conducted to examine the conformity revealed a five-factor structure. The five factors explained 39.8% of the total cumulative variance. Children with a low SES yielded lower scores in all subdomains except participation in physical activity. This effect was most pronounced in motor skills (*p* < 0.001, *r* = 0.28). Among the children with a low SES, 16.9% were overweight and 17.4% were obese, compared to 11.5% and 5.4%, respectively, for those with a high SES (*p* < 0.001). In conclusion, although the assessment tool was largely in line with the postulated structure, further adjustments are necessary in terms of participation and motivation. Nevertheless, this holistic view of PL, taking SES into account, should enable the focused promotion of health and health literacy.

## 1. Introduction

The benefits of exercise and physical activity in childhood and adolescence are undisputable [1]. Sufficiently active children exhibit better physical, mental, and spiritual health than their less-active peers [2]. However, the rates of physical activity, which had already decreased significantly in the wake of digitalisation and increased domestication, have been further reduced by the impact of the COVID-19 pandemic [3,4]. Even before the pandemic, only 22.4% of girls and 29.4% of boys in Germany aged 3–17 years attained the 60 min of physical activity per day recommended by the World Health Organisation [5], increasing the risk of developing obesity, motor deficits, or cardiometabolic risk factors [2,6]. Furthermore, behavioural patterns and health-related attitudes established in childhood often persist into adulthood [7].

Children with a migration background and/or a low socioeconomic status (SES) are particularly affected by a lack of physical activity and its potential consequences; for example, data from the Child and Adolescent Health Survey (KiGGS) revealed that girls and boys from families with a high social burden were significantly less likely to be active in sport outside of school than children from families with a low social burden [7]. Moreover, the former were at increased risk of overweight and obesity [8], making appropriate countermeasures for these children all the more important. However, no uniform and sustainably effective silver bullet has yet been established [9].

One possible approach in this context is the concept of physical literacy (PL), which is similar to the health literacy (HL) approach. HL [10], initially conceived in the 1970s, is essentially concerned with teaching skills for acquiring, evaluating, and (critically) applying health information [11]. Based on questionnaire surveys (e.g., the EU-HLS-47), children and families with higher HL are characterised by a correspondingly better health status [12]. Contrariwise, low HL is more likely to be associated with a lower SES; lower educational attainment; a less healthy lifestyle; and, as a result, poorer general health and higher mortality [13,14]. Schools, especially primary schools, offer an opportunity to address social and health inequalities, including those related to lifestyle and physical activity; however, the conditions—also in terms of sports equipment—are less favourable in disadvantaged neighbourhoods [15]. In this light, PL may support health-related behaviour, specifically physical activity, from a holistic perspective that regards the human being as a unity of body and mind (monism) or as the result of collected experiences in the world (existentialism), which in turn form the basis for one’s own process of perception (phenomenology) [16]. Working from this perspective, Margaret Whitehead developed the core components of motivation and confidence (affective dimension), physical competence (physical dimension), and knowledge and understanding (cognitive dimension). Therefore, the International Physical Literacy Association defines PL as the ‘motivation, confidence, physical competence, knowledge, and understanding to value and take responsibility for engagement in physical activities for life’ [17]. Even if this definition is widely accepted, it can be found internationally in different variants [18]. An internationally uniform definition is not yet available.

However, the abovementioned core components should not be viewed in isolation but as interacting and interdependent categories of a holistic construct that ‘individuals develop in order to maintain physical activity at an appropriate level throughout their life’ [19]. Thus, the foundation for lifelong participation in physical activity in the sense of the principle of ‘lifelong learning’ should already be laid in childhood, enabling the course to be set for positive health behaviour into adulthood [20]. Hence, while HL focuses mainly on ‘academic’ skills that can be used to apply information and resources that are necessary for maintaining and promoting health [21], the PL concept includes the aspects of (self)confidence, motivation, and movement-related knowledge and skills [22]. A focus on PL is essential in childhood, because ‘competencies’ such as confidence, motivation, and motor skills—as opposed to mere health-related knowledge—can be promoted in a meaningful, age-appropriate way. Due to this special importance, PL could be integrated as a more relevant building block in HL.

A prerequisite, in addition to a uniform definition, entails establishing nationally or internationally coordinated assessment tools with which PL can be measured. In childhood and adolescence, the ‘RBC Learn-to-Play-Canadian-Assessment-of-Physical-Literacy—Second Edition (CAPL-2)’, an assessment originally developed to monitor the PL of Canadian children aged 8 to 12 years [23], is typically used. This tool comprises 14 items (25 in the first edition) covering the subdomains of motor skills, knowledge, and understanding, as well as motivation and self-efficacy, summarised as the ‘affective’ domain [24]. In addition, the individual’s participation in physical activity is assessed as a fourth subdomain. Other PL assessment procedures for different target groups and settings have also been described in the international literature [25]. Examples include PrePLAY, which focuses on preschool children [26], and Sum et al.’s use of the Perceived Physical Literacy Instrument to assess the PL of physical education teachers [27]. To date, however, none of the available holistic PL assessments have considered primary school children from the age of 6 and up [28], even though, in this age group, health-promoting measures may be particularly crucial [29]. Therefore, to close this gap, we adjusted the Canadian model and operationalised the respective subdomains adapted from the literature (see Figure 1). From this, we constructed an assessment tool for children aged 6 years and above and extended the assessment procedure for the cognitive and affective domain to enable a differentiated consideration of PL.

Accordingly, the assessment was developed from the literature, and its construct validity was tested in primary school children while taking SES into account.

## 2. Materials and Methods

### 2.1. Study Population

Data collection took place between September and December 2021 in the form of a cross-sectional study across four primary schools in the city of Cologne. The recruited and selected schools were part of the STUPS project, a school- and community-based participatory approach for promoting physical activity in children and their families in disadvantaged neighbourhoods [31]. This project aims to develop a community-based approach to promoting physical activity by increasing physical literacy among elementary school children and their household members. Approval for the study was obtained from the ethics committee of the German Sport University Cologne (136/2019). Sickness or quarantine due to COVID-19 was considered the only exclusion criterion. The selection of schools took account of socioeconomic status (SES), meaning that schools from locations featuring both low and high social burden were included. Consent forms were distributed to all children in the participating schools, and all children who could provide signed consent from their parents took part.

A total of 567 children in grades 1 to 4 and aged 6 to 12 were recruited. Of these, 293 were male (51.7%), and the average age was 8.0 ± 1.3 (6 to 12) years. From the study sample, 437 children (77.1%) came from a total of three primary schools with a high social burden (=low SES), while 130 children (22.9%) came from one school characterised by a low social burden (=high SES).

#### Socioconomic Status Variable: School-Based Social Index

The social burden of the children was determined using the school-based index from North Rhine-Westphalia, Germany (‘Schulscharfer Index’) [32]. This index comprises the following four indicators:Child and youth poverty—measured by the proportion of children living near their primary school and whose parents receive state support for job-seekers (SGB II rate);The proportion of children in families whose mother tongue is not German;The proportion of children who have immigrated from abroad;The proportion of children with special needs in the areas of learning, emotional and social development, and language.

The classification is based on a social index scale of 1–9, with lower numbers indicating a lower social burden and higher numbers representing a higher social burden. In the following analysis, the social index was used as a dichotomous variable, with index levels 1–5 categorised as a low social burden (=high SES) and levels 6–9 as a high social burden (=low SES). The allocation was based on statistical data for the city of Cologne, with a social index defined for each city district comprising information on topics such as economic, political, and cultural disadvantages, as well as health inequality [33].

### 2.2. Anthropometric Data, Age, and Sex

Anthropometric data were collected from each child, while age and sex were self-reported by the children and compared with class lists. Height (cm) was measured using a calibrated stadiometer. During measurement, the participants were barefoot and in an upright posture while aligning the head with the Frankfurt horizontal. Body weight (kg), including clothing but without shoes, was also determined using standardised scales; meanwhile, abdominal circumference was measured using a tape measure according to WHO specifications [34,35].

After these physical measurements were taken, the individual body mass index (BMI kg/m^2^) was calculated. The values obtained were categorised based on percentile curves according to the German norm values of Kromeyer–Hauschild, classifying those who were above the 90th percentile as overweight and those who were above the 97th percentile as obese [36]. In addition, the BMI standard deviation score (SDS) was calculated using the least mean squares (LMS) method for non-normally distributed characteristics [37].
(1)$${\mathrm{SDS}}_{\mathrm{LMS}}=\frac{{\left(\frac{\mathrm{BMI}}{M\left[t\right]}\right)}^{L\left[t\right]}-1}{L\left[t\right]S\left[t\right]}$$
where BMI-SDS is the body mass index standard deviation score, which indicates the difference between an individual BMI value and the age- and sex-specific BMI median, and M[t], L[t], and S[t] are parameters for the age and sex of the test subjects.

### 2.3. Operationalisation of the PL Model and Assessment

The following subdomains were operationalised according to the Canadian model: (1) participation, (2) motivation and self-efficacy, (3) knowledge and understanding, and (4) motor skills. The dimensions of participation, motivation, self-efficacy, and knowledge were measured using specially designed questionnaires whose questions were developed from the literature and answered by the subjects in the form of a bipolar 6-point Likert scale with verbalised endpoints. The same assessment tool was used for all classes in age-adapted survey formats. First and second graders were interviewed in the form of an approximately 15 min one-to-one interview. The test administrators documented the children’s answers verbatim in the questionnaire. The children in the third and fourth grades, meanwhile, were given the questionnaire to complete independently under the guidance of a test leader. The test leader read the questions aloud, was available in case of comprehension difficulties, and checked the questionnaires for completeness afterwards. If a language barrier made it difficult for a participant to answer the questions independently, in individual cases involving older children, the survey was alternatively conducted in the form of an interview. The subdomain motor skills were tested using the Dordel–Koch test (DKT) in groups of 6–8 children [38,39].

All tests took place during school or childcare hours at the schools’ sports grounds. All test assistants received training in advance on how to conduct the examinations and exercises. The training session lasted a total of 90 min, during which the construct PL was explained to the assistants and the motor tests and questionnaire were presented. Afterwards, the interview was discussed in pairs, and the motor tests were carried out. The assistants were also given detailed written instructions on how to record and conduct the tests correctly and consistently.

#### 2.3.1. Participation: Physical Activity in Everyday Life and Leisure, Sport, and Sedentary Behaviour

The assessment of participation in physical activities was carried out in pictorial form as an ‘activity pyramid’ (Figure 2). This format was chosen to make the questions more accessible to the target group [40]. There were three intensity levels distinguished from each other: light (=active everyday life), moderate (=active leisure time), and intensive (=sport) [41]. This classification was designed to correspond to the metabolic equivalent (MET) intensity levels [42,43]. The respective categories contained a scale in the form of six increasing circles. The lowest, smallest circle corresponded to the lowest level of participation (=1 point), while the highest, largest circle corresponded to the most extensive participation (=6 points).

In addition, the extent of sedentary behaviour through the use of inactive means of transport in everyday life (=inactive everyday life), the pursuit of musical and artistic activities (=leisure time non-medial), and the consumption of digital media (=leisure time digital) was queried. The 6-point scale was also used in these categories (Figure 2). The points for sedentary behaviour were not taken into account in the overall evaluation of the PL but were instead integrated into the calculations as part of a preliminary evaluation.

#### 2.3.2. Motivation and Self-Efficacy

Motivation was assessed by recording specific motives (= item ‘motives’) or physical activity from participants’ responses to an open question, while the general enjoyment (= item ‘enjoyment’) of physical activity was recorded by means of a 6-point smiley analogue scale that ranged from 6 points for the happiest smiley (‘very, very happy’) to 1 point for the least happy smiley (‘not at all happy’; Figure 3).

The assessment of the open question was based on the main features of the self-determination theory proposed by Deci and Ryan [44]. This model distinguishes different qualities of motivation according to their degree of self-determination, forming a continuum that ranges from ‘no motivation’ to ‘controlled motivation’ and ‘autonomous motivation’. The assessment drew from the children’s answers to these questions regarding their motives for movement, which were initially clustered in an inductive procedure and then assigned to categories (see Table A1). A child whose response suggested an unmotivated attitude towards movement was assigned to the category ‘amotivated’ and received 1 point. ‘Controlled motivation’ included all types of other-determined and external motivation [45] (e.g., pressure from parents to perform), and 3.5 points were awarded for this category. ‘Autonomous motivation’ included all intrinsic motivation and increasingly self-determined extrinsic motivation; participants whose answers fell within this category had the highest degree of self-determination and were awarded 6 points [46]. The categories were assigned by an expert team of three researchers. Any controversies were resolved through discourse or by adding a fourth expert opinion.

Self-efficacy was assessed using two questions in a pictorial format. As in the case of participation, the procedure was intended to create an identification basis for the subjects from the ‘General Self-Efficacy Scale’ developed by Schwarzer et al. [47]. This questionnaire stems from the theoretical model of social-cognitive learning theory proposed by Bandura [48], the central element of which is the self-efficacy expectation, which was recorded in the item ‘confidence’ [49]. The example showed a fictitious wall that had to be overcome (Figure 4). Using a 6-point scale, the children had to indicate how much confidence they felt when facing a similar unfamiliar sporting challenge (not at all = 1 point; fully = 6 points). In terms of self-efficacy expectation, the aim was to test whether the child was convinced that they could consciously master or confront a new challenge [47]. The children were then asked how they would proceed if they did not master the wall or challenge (Figure 4). The scale ranged from 1 point (‘walk away and don’t try again’) to 6 points (‘practice and try again’).

#### 2.3.3. Knowledge and Understanding

The participants’ knowledge and understanding of the effects of physical activity were assessed via two items in the form of open questions. In the first part, the respondents were asked to describe in their own words how sport and physical activity affected their well-being and especially their feelings (= item ‘feelings’). In the second part, the children described what changes they perceived in their bodies during or after a sporting activity (= item ‘body‘). In this case, a distinction was made as to whether the answers referred to long- or short-term effects (= item ‘time span’) [50]. The answers were inductively clustered by three experts and assigned to generic terms; in cases of ambiguity, a fourth expert opinion was consulted (Table 1). The respective answers that were assigned to an umbrella term can be seen in Table A2.

A maximum of three answers each from the sub-areas ‘feelings’, ‘body’, and ‘time span’ were scored. For the latter, points were only awarded when the child named long-term effects on the body; short-term effects did not receive any points. Each child received a flat 1 point per category. For example, if a child’s answer included ‘happy, sweat, lose weight, get stronger’, they would be awarded a flat 3 points plus 1 point for feelings (‘happy’), 3 points for body (‘sweat, lose weight, get stronger’), and 2 points for time span (‘lose weight, get stronger’).

#### 2.3.4. Motor Skills

Motor skills were assessed using selected items from the DKT. From the seven test items, the lateral jumping, standing long jump, and 6 min run were recorded. The three selected test items are all well validated, and some of them are also used in other motor skills tests [38,39].

Agility and coordination were assessed using the lateral jumping item. For this purpose, lateral jumps in which both legs crossed a line at the same time were performed in two rounds of 15 s each, and the number of jumps from both rounds was added together. The strength of the lower extremities was determined via the standing long jump. For this purpose, the best of two two-legged jumps from a standing position was measured in centimetres. Lastly, cardiorespiratory and aerobic endurance performance was measured by having the participants complete a 6 min run in which they attempted to cover as many metres as possible [51]. The 6 min run correlates with VO_2_max; the assessment of additional parameters, e.g., heart rate for more precise results, did not take place. A detailed description of the complete test battery, as well as background information on the definition of motor performance, is available in the literature [38,52].

The sports motor tests were evaluated using the scoring system of the DKT, which includes scoring tables for the individual test items according to age- and gender-specific standardised data. The evaluation takes the form of school grades with inverse scoring (grade ‘very good’ = 6 points, grade ‘good’ = 5 points, etc.).

#### 2.3.5. Total Scoring

Figure 5 presents the distribution of points. The final score was made up of 18 points each for the subdomains of motor skills, participation, and motivation/self-efficacy and 6 points for the subdomain of knowledge and understanding. The weighting of the subdomain knowledge and understanding arose from the results of a Delphi process for the development of the Canadian Assessment of Physical Literacy (CAPL) [24,53].

Thus, a maximum of 60 points could be achieved in the overall assessment of PL. The scoring system was also developed from the explorative factor analysis that we conducted. Given that the three inactive items of the activity pyramid (inactive everyday life, leisure time non-medial, and leisure time digital) could not be summarised effectively in a single factor, they were not given further consideration in the scoring.

### 2.4. Statistical Analysis

The data were analysed using the statistical program IBM SPSS version 28.0. The frequencies, mean values, and standard deviations of the anthropometric data are presented separately according to sex and SES. Differences in the individual parameters were calculated using the *t*-test for independent samples. Testing for differences in the frequencies of BMI percentiles and sex or SES was accomplished by the chi-square test for independence. The significance level was set at *p* < 0.05. Descriptive statistics were also used for the individual items of the PL subdomains (see Table A3).

Dimensionality and construct validity were tested through an exploratory factor analysis that only included subjects whose dataset was complete (n = 541), which meant that the results from 26 children had to be excluded.

The suitability of the data was determined using the Kaiser–Meyer–Olkin criterion (KMO) and the measure of sampling adequacy (MSA). The number of extracted factors was determined using Cattell’s Scree Plot [54]. The threshold value for factor loading was set at 0.5, and that for the minimum eigenvalue of the factor loading at 0.3 [55]. In addition, the item difficulties were calculated.

Furthermore, a Mann–Whitney U test was used to determine whether the achieved scores in the subdomains differed across the status groups. For this test, the significance level was set at *p* < 0.05, the effect size was calculated with the correlation coefficient *r*, and the *r*-value was interpreted according to Cohen as follows: *r*~0.1 = weak effect; *r*~0.3 = moderate effect; *r*~0.5 = strong effect [56].

## 3. Results

### 3.1. Anthropometric and Socio-Demographic Data

The complete set of anthropometric data is presented in Table 2. Statistically significant differences between the sexes were found for waist circumference (*t*(565) = 2.3, *p* = 0.021) and BMI-SDS (*t*(565) = 2.15, *p* < 0.032). In addition, boys were found to be more likely to be overweight and obese, while girls were more likely to be underweight (χ^2^(3) = 8.49, *p* = 0.037).

### 3.2. Exploratory Factor Analysis

In the anti-image correlation, the MSA showed a value > 0.5 for all items except ‘inactive everyday life’ (=0.493). This factor was therefore excluded from further analysis and no longer considered. The MSA values of all other items ranged from 0.529 to 0.786. The KMO (=0.614) yielded a value >0.5 for the sample adequacy for all items and was thus suitable for the factor analysis. Bartlett’s test (chi-square χ2 = 1450.550; df = 105) confirmed the items’ correlation with each other (*p*-value < 0.001). Maximum likelihood analysis (MLE) with varimax rotation revealed a factor structure of five factors with eigenvalues > 1.0, which was confirmed by the scree plot with Kaiser normalisation. In addition, extracted communalities ranged from 0.036 to 0.999, and item difficulties ranged from 26.6 to 88.

Five factors explained 39.8% of the total cumulative variance. Two items loaded on factor 1, three items on factor 2, and two items on factor 3. Factor 4 contained one item, and factor 5 contained three items. The resulting factors yielded the following five scales (Table 3):Scale 1: Knowledge of body and time span (E1, E2);Scale 2: Motor skills (A1, A2, A3);Scale 3: Self-efficacy (D1, D2);Scale 4: Knowledge about feelings (E3);Scale 5: Participation and motivation (B2, B3, C2).

Only four items had factor loadings below 0.3 and could therefore not be assigned to any factor.

### 3.3. SES and Anthropometric Data

Children assigned to the high-burden category underwent an average increase in BMI-SDS of 0.46 (95% CI [0.24, 0.68]), *t*(565) = 4.12, *p* < 0.001. Waist circumference was also significantly larger by an average of 2.71 cm (95% CI [0.87, 4.56]) in children assigned to the high-burden category, *t*(565) = 2.89, *p* = 0.002.

The distribution of the two groups in terms of percentiles differed significantly, χ^2^(3) = 16.41, *p* < 0.001. The proportion of overweight and obese children was 16.9% and 17.4%, respectively, for low SES, which was higher than the respective 11.5% and 5.4% for high SES. In contrast, underweight was described significantly more often in the group with high SES (6.2 vs. 3.9%). All data are displayed in Table 4.

### 3.4. SES and PL Subdomains

Significant differences emerged between the two groups in all PL subdomains and the total score (summarised in Table 5). The children with higher SES scored higher in almost all domains. The strongest effect was found in motor skills, *U* = 17,131.00, *Z* = 6.585, *p* < 0.001, *r* = 0.28. In addition, the low-burden group achieved higher values in the subdomains of motivation and self-efficacy, *U* = 23,607.00, *Z* = 2. 516, *p* = 0.012, *r* = 0.11; knowledge and understanding, *U* = 20,590.00, *Z* = 4.779, *p* < 0.001, *r* = 0.20; and the overall PL score, *U* = 22,947.00, *Z* = 2.242, *p* = 0.025, r = 0.1. Only in regard to participation did the subjects in the group with low SES exhibit higher values on average: *U* = 19,327.00, *Z* = 5.239, *p* < 0.001, *r* = 0.22.

## 4. Discussion

To the best of our knowledge, this study was the first investigation and development of a model-based assessment tool for PL in primary-school-age children in Germany whose (clinical) relevance was tested in relation to SES. The explorative factor analyses showed good preliminary results for the developed PL assessment. Significant differences in the PL domains of motivation and self-efficacy, motor skills, and knowledge and understanding were found in favour of the children with a low social burden. The differences were particularly pronounced in motor skills. Only in participation in physical activity did the children with a high social burden achieve significantly higher scores.

### 4.1. Content Validity

The operationalisation was preceded by a literature-based examination of the topic of PL. The aim was to present the construct as precisely as possible in terms of content validity, requiring existing assessment tools, such as the CAPL [24], to derive the subdomain-specific items. However, the approaches available up to this point have been characterised by inconsistent content and structural assessment systems or definitions [25,57]. This phenomenon is most apparent in the cognitive subdomain: thus, in the literature, as well as in this paper, this subdomain is mostly understood as the knowledge and understanding of the positive effects of sport and exercise [24,58]. Nevertheless, George et al. [59] also recorded intrinsic motivation (affective domain) within the cognitive domain, while Rudd et al. [60], among others, identified executive functions. In contrast, the CAPL-2 uses formative indicators to query a participant’s learning level on the topic of sport and exercise [61]. During development, therefore, we started exploratively from the knowledge of the target group itself. This approach facilitated an unbiased mapping of children’s assessments of the effects of sports and physical activity. However, the extent to which this method reliably depicts the subdomain must be analysed in further research to confirm the measurement accuracy of the subdomains.

### 4.2. Exploratory Factor Analysis

The study’s exploratory factor analysis allowed the essential factors of the underlying hypothetical model of PL to be described.

The motor function domain consistently indicated factor loadings above the threshold value of 0.3. Strength, endurance, and coordination/mobility were interrelated: good performance in one skill correlated with good performance in all of the other motor skills [62]. This finding was in line with other studies’ confirmation of correlations between and within dimensions. Similarly to our model, Pastor-Cisneros et al. demonstrated that physical competence, motivation and self-efficacy, and self-perceived fitness positively influenced each other [63].

The factors in the subdomain of motivation and self-efficacy were not entirely exact. While the two items of the self-efficacy domain (‘confidence’ and ‘practice and retry’) loaded on a common factor, the items for motivation were distinct from that factor. This finding was in line with the International Physical Literacy Association’s definition of motivation and self-efficacy, which treats these two elements as separate subdomains [28]. Furthermore, the item ‘motives’ for physical activity did not load on any of the five factors. A possible explanation for this outcome could be related to the three-level scaling (amotivated, controlled motivated, autonomously motivated), which naturally allowed little variance. In order to be able to depict movement motives within the assessment tool, we would recommend using a scale for the item that has at least five or, preferably, six levels. The extent to which the model used in the current study needs to be adapted accordingly remains the subject of further research.

In the case of participation, the item ‘active everyday life’ did not load high on any factor, in contrast to the other two movement-related indicators from the activity pyramid, which could be assigned to a factor. In general, the recording of participation via a self-report questionnaire is methodologically challenging. Questionnaire formats are less reliable and are often influenced by external factors, such as social desirability, fluctuations, or the complexity of the questionnaire [64]. Furthermore, it is particularly difficult for children to recall their exact physical-activity-related behaviour over longer periods of time [65]. Thus, in follow-up studies, the ‘Previous Day Physical Activity Recall (PDPAR)’, a well-studied and frequently used self-assessment questionnaire, could be used because it only refers to the previous day [66]. Another option would be to add an objective measuring device (accelerometer, pedometer), as is also provided for in the CAPL [24]. In the literature, this method is considered particularly suitable for younger children; however, it also comes with the drawbacks of being more time-consuming and financially costly [67,68].

### 4.3. Impact of SES

In the subdomains of motor skills, motivation and self-efficacy, and knowledge, children from primary schools with a high social burden demonstrated worse performance. Particularly notable, however, was that we also found the highest proportion of overweight and obese children in these neighbourhoods. In previous investigations, Delisle Nyström et al. and Comeau et al. examined whether there existed an association between PL subdomain scores and the weight status of 8–12-year-olds [69,70]. Their studies also showed that overweight and obese children scored lower in all four domains and in their overall score. In general, however, it must be assumed that the common denominator is the impeded access to a healthy lifestyle caused by factors such as financial hardship, a lack of physical activity, and fewer opportunities for education and the promotion of appropriate health behaviours [71]. These factors also often imply motor and sports-related deficits [72]. On the other hand, a higher percentage of underweight children was described in the group with a lower social burden. The extent to which this indicates a possible health concern in the sense of an eating disorder at primary school age remains a matter of speculation.

Surprisingly, though, children from primary schools with a higher social burden scored better in the context of participation, contradicting many studies that show at least an inverse correlation in club activity [73]. The extent to which the methodological approach may have led to a misjudgement in this work is a matter of speculation at present and will be re-examined in follow-up studies through the potential additions that were previously mentioned.

### 4.4. Strengths und Weaknesses

This study is one of the first to record the PL of primary school children in Germany. In general, a sample size of 541 subjects is judged in the literature to be a ‘very good’ prerequisite for conducting an exploratory factor analysis [74]. Therefore, no sample size calculation was performed for the analysis of the PL subdomains according to SES.

The bottom-up approach (i.e., developing clusters participatively and inductively via the target group survey) was a particular strength of this study.

However, due to partially low-level response scaling (e.g., subdomain motivation), statistical suitability could only be generated to a limited extent on the basis of the communalities. Nevertheless, the other prerequisite tests (anti-image correlation, Bartlett test, KMO) confirmed that the data were suitable for explorative factor analysis. Ultimately, the five-factor structure proved to be a satisfactory preliminary model for gaining initial insights into the underlying construct. The next step will be to further develop the construct validity of the assessment tool through supplementary procedures, especially in the context of motivation and participation. In addition, the assessment‘s reliability and measurement accuracy must be tested in future research.

At the same time, it should be emphasised that the suitability of our test method was verified in a field test. However, it should be noted that the children’s SES was not assessed individually but via the School-Based Social Index. Although this procedure has also been implemented in comparable studies [75], it might not have been appropriate in individual cases. However, the extent to which this approach actually influenced the outcomes remains a matter of speculation. In order to be able to reflect the relationship between social inequality more precisely, subsequent studies will examine the SES of families individually, e.g., by using the Family Affluence Scale [76].

## 5. Conclusions

In general, the assessment tool we presented is promising for the adequate recording and assessment of PL in 6- to 12-year-olds. As a next step, the weaknesses we discussed will be eliminated, and the new adaptations will be evaluated on the basis of further tests in different target groups, thus contributing towards the goal of establishing a universally applicable assessment for recording individual PL. In the long term, the goal is to develop the means to identify problem areas to enable needs-oriented action in the context of physical activity and health promotion at an early stage. Furthermore, such approaches can be used to promote children’s (and young people’s) health from a holistic perspective. This is particularly true for children from families with a lower level of education. Children with a low SES and migration background were more likely to exhibit unfavourable health behaviour patterns, poorer motor skills, and higher BMI scores [77]. In children and adolescents, PA accumulated in the neighborhood, school, and recreational environment is examined most frequently [78]. To increase PA and to foster PL, parks, playgrounds, and other green spaces should be made safe, attractive, and activity-friendly, and transport infrastructure that is pedestrian and cycling friendly should be provided. This also applies to (primary) schools, which should not only be well- or better-equipped for physical education lessons, but also make an important contribution to the promotion of PL through physical activity breaks and after-school programs. In line with the concept of a “health literacy environment”, future research and practice should focus not only on PL at an individual and population level, but also on the surrounding environment where children (and youth) grow up.

## Figures and Tables

**Figure 1 children-09-01908-f001:**
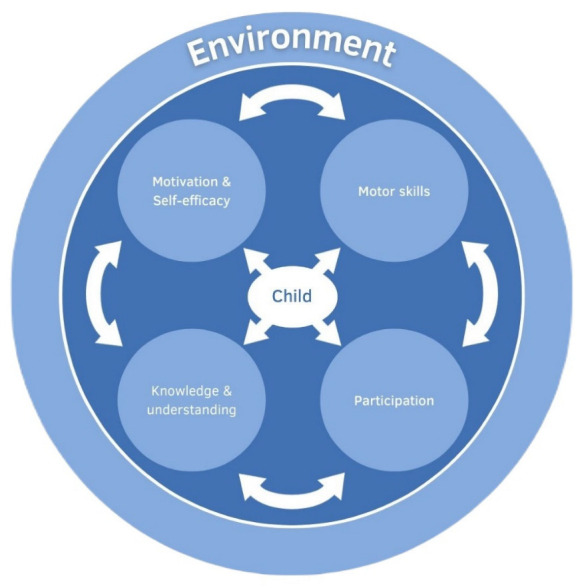
Theoretical model as a basis for the assessment tool [30].

**Figure 2 children-09-01908-f002:**
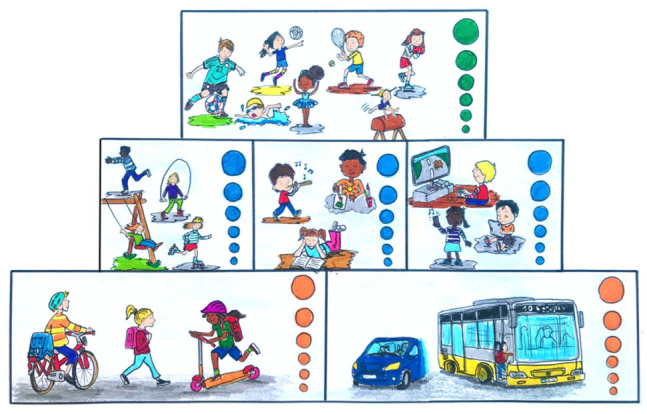
Activity pyramid to measure participation: lower level—everyday life (left active = light intensity, right inactive); middle level—leisure time (left active = moderate intensity, middle non-medial, right digital); upper level—sports = high intensity.

**Figure 3 children-09-01908-f003:**
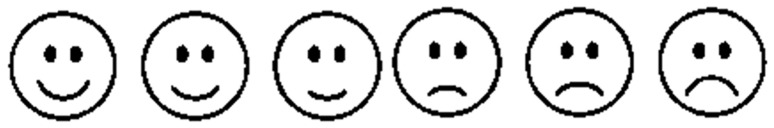
6-point smiley analogue scale, ranging from the happiest (‘very, very happy’) to the saddest smiley (‘not at all happy’).

**Figure 4 children-09-01908-f004:**
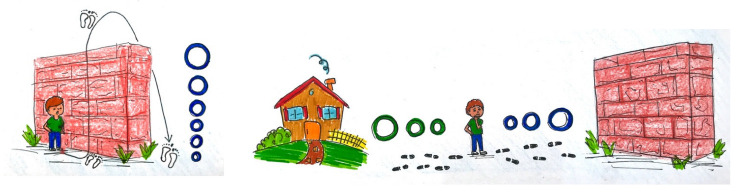
‘Confidence’ and ‘practice and try again’ items to measure self-efficacy.

**Figure 5 children-09-01908-f005:**
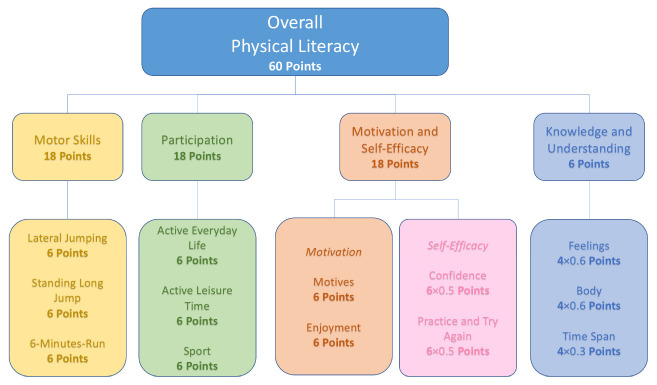
Scoring of the subdomains and weighted composition of the overall assessment of PL; the maximum achievable score is shown for each case.

**Table 1 children-09-01908-t001:** Examples of the clusters from the areas of knowledge related to ‘feelings’ and ‘body, long- and short-term (time span)’.

Feelings	Body	
	Short-Term Effects	Long-Term Effects
relaxation	physical strain	endurance
stress relief	lungs/breathing	flexibility
self-confidence	cardiovascular system	coordination
fun	side stitches	speed
joy	muscle ache	health
strength		learning to move
courage		weight loss
		fitness
		gaining muscles

The answers of the individual headings are listed in Table A2.

**Table 2 children-09-01908-t002:** Descriptive statistics of anthropometric data broken down by sex.

Parameter	Total		Girls		Boys		*p*-Value
	*n*	Mean ± SD	*n*	Mean ± SD	*n*	Mean ± SD	
age (years)	567	8.0 ± 1.3	274	8.0 ± 1.2	293	8.0 ± 1.3	0.780 ^+^
height (cm)	567	129.7 ± 9.3	274	129.2 ± 9.0	293	130.1 ± 9.5	0.286 ^+^
weight (kg)	567	30.5 ± 8.6	274	29.9 ± 8.6	293	31.0 ± 8.7	0.110 ^+^
waist (cm)	567	61.8 ± 9.5	274	60.9 ± 9.3	293	62.7 ± 9.5	0.021 ^+^
BMI (kg/m^2^)	567	17.9 ± 3.4	274	17.6 ± 3.3	293	18.1 ± 3.4	0.093 ^+^
BMI-SDS	567	0.64 ± 1.14	274	0.53 ± 1.14	293	0.74 ± 1.13	0.032 ^+^
percentile %	567		274		293		0.037 ^++^
underweight	25	4.4%	18	6.6%	7	2.4%	
normal weight	370	65.3%	177	64.6%	193	65.9%	
overweight	89	15.7%	46	16.8%	43	14.7%	
obese	83	14.6%	33	12.0%	50	17.1%	

Abbreviations: *n* = number; BMI = body mass index; SDS = standard deviation score; *p*-value < 0.05 = significant; ^+^ = calculation with *t*-test; ^++^ = calculation with chi-square test.

**Table 3 children-09-01908-t003:** Rotated factor matrix of the exploratory factor analysis.

Item	Factor					Item Difficulty
	1	2	3	4	5	
A1 lateral jumping		0.633				47.6
A2 standing long jump		0.607				37.6
A3 6 min run		0.580				31.2
B1 active everyday life						73.2
B2 active leisure time					0.501	76.6
B3 sport					0.462	81.6
B4 leisure time non-medial						70.2
B5 leisure time digital						69.6
C1 motives						88.0
C2 enjoyment					0.420	92.8
D1 confidence			0.593			71.8
D2 practice and try again			0.775			73.4
E1 body	0.958					27.7
E2 time span	0.889					37.0
E3 feelings				0.992		26.6

Factor loading ranged from −1.0 to 1.0; absolute values closer to 1.0 represented a closer link to the observed item; the threshold of factor loading in this study was 0.30.

**Table 4 children-09-01908-t004:** Descriptive statistics of anthropometric data broken down by SES.

Parameter	Low SES		High SES		*p*-Value
	*n*	Mean ± SD	*n*	Mean ± SD	
female sex %	215	49.2%	59	45.4%	
male sex %	222	50.8%	71	44.6%	0.445 ^++^
age (years)	437	8.0 ± 1.3	130	8.3 ± 1.3	0.005 ^+^
height (cm)	437	128.9 ± 9.3	130	132.2 ± 8.9	<0.001 ^+^
weight (kg)	437	30.7 ± 9.1	130	29.7 ± 6.6	0.094 ^+^
waist (cm)	437	62.4 ± 9.9	130	59.7 ± 7.5	0.002 ^+^
BMI (kg/m^2^)	437	18.2 ± 3.6	130	16.9 ± 2.3	<0.001 ^+^
BMI-SDS	437	0.75 ± 1.17	130	0.28 ± 0.93	<0.001 ^+^
percentile %	437		130		<0.001 ^++^
underweight	17	3.9%	8	6.2%	
normal weight	270	61.8%	100	76.9%	
overweight	74	16.9%	15	11.5%	
obese	76	17.4%	7	5.4%	

Abbreviations: SES = socioeconomic status; higher SES (indicating low social burden) = Social Index 1–5; lower SES (indicating high social burden) = Social Index 6–9; *n* = number; BMI = body mass index; SDS = standard deviation score; *p*-value < 0.05 = significant; ^+^ = calculation with *t*-test; ^++^ = calculation with chi-square test.

**Table 5 children-09-01908-t005:** SES and PL subdomains.

Parameter	Low SES		High SES		*p*-Value	*r*-Value
	*n*	Median (IQR)	*n*	Median (IQR)		
Motor skills	432	8 (3)	128	10 (4)	<0.001	0.28
Participation	429	15 (4)	129	13 (4)	<0.001	0.22
Motivation and self-efficacy	424	16.5 (4.44)	130	17 (2.5)	0.012	0.11
Knowledge and understanding	436	2.7 (1.2)	130	3.3 (0.9)	<0.001	0.20
Overall physical literacy	416	42.5 (7.4)	127	43.2 (8)	0.025	0.10

Abbreviations: SES = socioeconomic status; higher SES (indicating low social burden) = Social Index 1–5; Lower SES (indicating high social burden) = Social Index 6–9; *n* = number; IQR = interquartile range; *p* < 0.05 = significant; *r* = correlation coefficient (*r*~0.1 = weak, *r*~0.3 = moderate, *r*~0.5 = strong.

## Data Availability

The data presented in this study are available on reasonable request from the corresponding author. The data are not publicly available due to data protection reasons.

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
