# Peer review of "Development of a German Physical Literacy Assessment for Children in the Context of Health Promotion—An Explorative Approach"

_children, 2022, doi:10.3390/children9121908_

Round 1

Reviewer 1 Report

Comment: The introduction in general is well written and a gap in the literature is presented.

Comment: In the methods, it is necessary to make it clearer how the recruitment was. Were schools selected at random or for convenience? How was the dissemination of the project? How were students and their parents informed of the purpose of the study?

Comment: The authors did not present the sample size calculation in the project. Did the sample size of 567 students have enough statistical power to answer the question of the present study?

Comment: Were the questionnaires applied in the present study validated? Was any type of reproducibility performed on a portion of the sample? For example, a small portion of the sample answered the questionnaire twice, with an interval of one week between one answer and another to analyze the agreement of the answers.

“All test assistants received training in advance on how to conduct the examinations and exercises”.

Comment: What was that kind of training like?

“Lastly, cardiorespiratory and aerobic endurance performance was measured by having the participants complete a 6-minute run in which they attempted to cover as many meters as possible”.

Comment: In this type of test, was there any care taken so that the children could not underestimate the test? For example, heart rate measurement after taking the test?

Comment: In the first paragraph of the discussion, please insert the main findings of the study.

“Pastor-Cisneros et al. demonstrated that physical com petence, motivation and self-efficacy, and self-perceived fitness positively influenced each other [59].

Comment: Why would this occur, what would be the mechanisms?

Reviewer 2 Report

Development of a German Physical Literacy Assessment for
Children in terms of Health Promotion – an explorative Approach

General Comment: There have been several studies of PL in recent times. This one explored use of a new assessment tool. The components are reasonable and the design appropriate. I like the idea of adding the SES component; one that would appear to be quite relevant and quite applicable to the German education system and physical education.  

Although the tool needs further work, the preliminary report is worthy of presenting to the research community; that is, with some clarifications needed.

One of the main pts of weakness is the relation of ‘Participation” (I assume Health Promotion) to Physical Literacy.. it’s a bit vague. How does it fit in? I understand after reading the Methods, but not earlier.

Specific Comments:

Abstract-

see comments on spelling.

Children with a low SES yielded lower scores in all subdo-17 mains except participation”          Where is the factor (variable) participation – not described earlier. Is it one of the 5     factors?

Introduction –

Good background literature review and support for specific context study (Germany).

“From this foundation, we constructed an assessment tool including an assessment scheme to       enable a differentiated consideration of PL in German children aged 6–12 years.” I would add “That is,…” how specifically does it differentiate German children from Canadian?  By SES alone? No. Somewhere in this concluding section on intent you need to insert the ‘participation’ variable.

Methods –

Its good to see a study looking at children from a community of high social burden (I like the       term).

“2.3.1. Participation: Physical Activity in Everyday Life and Leisure, Sport and Sedentary Behaviour” This        is the part that missing in the Abstract.

“The subdomain motor skills were tested using the 163 Dordel-Koch test (DKT) in groups of 6–8            children.” Reference?

Since context (SES) is a major factor in this contextual, study it would appear relevant to briefly describe the school physical education / sport programs at the different schools?     Typically, schools of lower SES have are at a disadvantage in regard to professional        instruction and facilities.

Reviewer 3 Report

Authors presents and interesting an innovative work focused in primary education, which is less common in literature. Is this sense, this is an important strength. Moreover, they made the intellectual effort of adapting the theoretical and practical approach of the CAPL-2 to children, which is also worthy. Nonetheless, it is necessary to consider that physical literacy is a complex concept and this reviewer is not sure if, for example it can be clearly differentiate among health and physical literacy. Therefore, as a theoretical and methodological concept I consider this is a meritorious manuscript that could add some insights to the physical activity and health-related literature framework, but I am not very sure about the applicability and implications of the result of this exploratory analysis.

Some other concerns are necessary to attend before of consider the manuscript for definitive publication.

SES variable: the SES scale only differs among low and high social burden. Although this is correct and common, I consider that the distribution of the sample is unbalanced toward low social burden. Is it possible that middle income contexts could be categorized as low. There are other scales, such as the Family Affluence Scale (Curio & Molcho) that consider the middle-income SES. Then, probably could be considered for further research.

Motivation and Self-Efficacy Scale: although authors explain the procedure to assess this domine, I do recommend to add a figure with the scale as they do with participation or confidence.

Weight status comparisons: authors highlight differences on overweight and obesity among SES, but they do not say something about underweight, which is higher for high SES and this could be considered a health-related concern, doesn’t it?

“problem” with participation: although as a theoretical and exploratory model it can be improved, this reviewer thinks that the results of participation does necessarily be a problem. Despite the potential contradiction with literature, it is also true that participation during childhood in Physical Activity is higher that, for example, adolescence. Therefore, Physical Literacy is more than participation.

Typing problems:

Page 11. Change Pas-tor-Cisneros for Pastor-Cisneros.

In conclusion, I do encourage author to rethink this concerns in order to make the final decision.

Round 2

Reviewer 1 Report

Dear Editor, all comments have been answered point by point and suggestions have been added to the manuscript. Therefore, I have no further comments to make.

Author Response

Thank you very much!